Identification of key genes as potential diagnostic biomarkers in sepsis by bioinformatics analysis

Lin Guoxin 1
Li Nannan 2 3
Liu Jishi 2 3
Sun Jian 2 3
Zhang Hao 2 3
Gui Ming 2 3
Zeng Youjie zengyoujie1995@gmail.com 1
Tang Juan csutj880109@163.com 2 3
1 Department of Anesthesiology, The Third Xiangya Hospital , Changsha , China
2 Department of Nephrology, The Third Xiangya Hospital, Central South University , Changsha , China
3 Clinical Research Center For Critical Kidney Disease In Hunan Province , Changsha , China
Zhang Xin
Electronic publication date: 2024 Jun 18
Publication date: 2024
Volume: 12
Electronic Location ID: e17542
Received 2023 Jul 27; Accepted 2024 May 19
Copyright: ©2024 Lin et al.
Copyright year: 2024
Copyright holder: Lin et al.
License: This is an open access article distributed under the terms of the Creative Commons Attribution License, which permits unrestricted use, distribution, reproduction and adaptation in any medium and for any purpose provided that it is properly attributed. For attribution, the original author(s), title, publication source (PeerJ) and either DOI or URL of the article must be cited.
License URL: https://creativecommons.org/licenses/by/4.0/

Keywords: Sepsis, Critical illness, Bioinformatics, Differentially expressed genes, Key gene, Biomarker

Funding: The Changsha Natural Science Foundation No. kq2208356 The National Natural Science Foundation for Distinguished Young Scholars of China 81900634 The Natural Sciences Foundation of Hunan Province for Distinguished Young Scholars No. 2021JJ40947 The National Natural Science Foundation of China No. 82070738 82270750 The Natural Science Foundation of Hunan Province No. 2021JJ31015 The Health Commision Fund of Hunan Province No. 202103050563 No. 202104022248 Hunan Province Clinical Medical Technology Innovation Guidance Project No. 2021SK53708 This work was supported by the Changsha Natural Science Foundation (No. kq2208356 to Jishi Liu), the National Natural Science Foundation for Distinguished Young Scholars of China (No. 81900634 to Juan Tang), the Natural Sciences Foundation of Hunan Province for Distinguished Young Scholars (No. 2021JJ40947 to Juan Tang), the National Natural Science Foundation of China (No. 82070738 and 82270750 to Jishi Liu), the Natural Science Foundation of Hunan Province (No. 2021JJ31015 to Jishi Liu), the Health Commision Fund of Hunan Province (No. 202103050563 and No. 202104022248 to Jishi Liu), Hunan Province Clinical Medical Technology Innovation Guidance Project (No. 2021SK53708 to Jishi Liu). The funders had no role in study design, data collection and analysis, decision to publish, or preparation of the manuscript.

==============================
Background

Sepsis, an infection-triggered inflammatory syndrome, poses a global clinical challenge with limited therapeutic options. Our study is designed to identify potential diagnostic biomarkers of sepsis onset in critically ill patients by bioinformatics analysis.

Methods

Gene expression profiles of GSE28750 and GSE74224 were obtained from the Gene Expression Omnibus (GEO) database. These datasets were merged, normalized and de-batched. Weighted gene co-expression network analysis (WGCNA) was performed and the gene modules most associated with sepsis were identified as key modules. Functional enrichment analysis of the key module genes was then conducted. Moreover, differentially expressed gene (DEG) analysis was conducted by the “limma” R package. Protein-protein interaction (PPI) network was created using STRING and Cytoscape, and PPI hub genes were identified with the cytoHubba plugin. The PPI hub genes overlapping with the genes in key modules of WGCNA were determined to be the sepsis-related key genes. Subsequently, the key overlapping genes were validated in an external independent dataset and sepsis patients recruited in our hospital. In addition, CIBERSORT analysis evaluated immune cell infiltration and its correlation with key genes.

Results

By WGCNA, the greenyellow module showed the highest positive correlation with sepsis (0.7, p = 2e − 19). 293 DEGs were identified in the merged datasets. The PPI network was created, and the CytoHubba was used to calculate the top 20 genes based on four algorithms (Degree, EPC, MCC, and MNC). Ultimately, LTF, LCN2, ELANE, MPO and CEACAM8 were identified as key overlapping genes as they appeared in the PPI hub genes and the key module genes of WGCNA. These sepsis-related key genes were validated in an independent external dataset (GSE131761) and sepsis patients recruited in our hospital. Additionally, the immune infiltration profiles differed significantly between sepsis and non-sepsis critical illness groups. Correlations between immune cells and these five key genes were assessed, revealing that plasma cells, macrophages M0, monocytes, T cells regulatory, eosinophils and NK cells resting were simultaneously and significantly associated with more than two key genes.

Conclusion

This study suggests a critical role of LTF, LCN2, ELANE, MPO and CEACAM8 in sepsis and may provide potential diagnostic biomarkers and therapeutic targets for the treatment of sepsis.

Introduction

As per the 2016 revision of the sepsis concept, sepsis is defined as a serious, life-threatening condition characterized by organ dysfunction resulting from an imbalanced response to infection (Singer et al., 2016). It represents the predominant cause for admission to intensive care units worldwide (Esposito et al., 2017). Epidemiological data from high-income countries suggest that over 31 million individuals are hospitalized for sepsis globally, with an estimated mortality of 5.3 million (Fleischmann et al., 2016). It is often difficult for clinicians to distinguish patients in the early stages of sepsis from critically ill patients with infection-negative systemic inflammation. Incorrect distinction and diagnosis can cause inappropriate patient management, overuse of antibiotics and result in death in severe cases (Prescott & Angus, 2018).

The current gold standard for the diagnosis of sepsis is the culture and identification of the causative pathogen, but this method may take more than 24 hours to obtain results (Evans et al., 2021). It has been reported that positive cultures are produced from only about 30% of clinically confirmed sepsis patients, without a clear explanation for negative culture results (Coburn et al., 2012). Assessment of the host immune response using biomarkers may provide a faster and more accurate way to differentiate sepsis from infection-negative systemic inflammation (Reinhart et al., 2012). One of the most studied biomarkers of host response is blood procalcitonin (Uzzan et al., 2006). However, there is increasing evidence that procalcitonin cannot provide a definitive diagnosis (Hoeboer et al., 2015). Single biomarkers in the blood have certain limitations due to the inherent complexity of the host immune response (Póvoa et al., 2023). Therefore, the identification of novel biomarkers from the sera of critically ill patients could provide an important tool for the early diagnosis of sepsis.

Bioinformatics analysis currently represents a widely employed methodology for screening massive gene expression datasets and plays a crucial role by identifying and analyzing differentially expressed genes (DEGs) associated with diseases (Ditty et al., 2010). In this study, we obtained the expression data of sepsis and non-sepsis critically ill patients from the Gene Expression Omnibus (GEO) database, with the specific objective of ascertaining early biomarkers of sepsis in critically ill patients. Initially, we conducted weighted gene co-expression network analysis (WGCNA) within datasets (GSE28750 and GSE74224) to pinpoint key module genes most associated with sepsis. Functional enrichment analysis of the key module genes was performed. Then, we identified DEGs of sepsis across the two datasets and subsequently constructed the PPI network with the STRING database, and discerned hub genes employing the cytoHubba app in Cytoscape software. Five key genes were obtained by intersecting the hub PPI genes and the key module genes of WGCNA. Subsequently, these key overlapping genes were validated in an independent external dataset (GSE131761) and sepsis patients recruited in our hospital. Additionally, the immune infiltration of sepsis was evaluated through CIBERSORT analysis and the correlation of key crosstalk genes with immune cells was calculated. Our research thus provides novel insights into potential mechanism and diagnosis avenues for sepsis in critical patients.

Methods

Dataset acquisition and combination

Sepsis-related gene expression profile datasets were obtained from the Gene Expression Omnibus (GEO) database (Clough & Barrett, 2016). The GSE28750 and GSE74224 datasets were used for exploratory analyses. The GSE28750 dataset consisted of 10 sepsis cases and 11 critically ill controls. The GSE74224 dataset included 74 sepsis cases and 31 critical illness controls. Table S1 showed the overview of sepsis datasets used in this study. Both datasets compared ICU sepsis patients and post-surgical patients with infection-negative systemic inflammation (McHugh et al., 2015; Sutherland et al., 2011). The “removeBatchEffect” function of the “limma” R package was used to remove the batch effect of the two datasets, thus merging the two datasets into one integrated gene expression dataset (Ritchie et al., 2015). All subsequent analyses were based on the merged dataset.

Weighted gene co-expression network analysis

Weighted gene co-expression network analysis (WGCNA) was performed to identify gene modules highly associated with sepsis by utilizing “WGCNA” R package (Langfelder & Horvath, 2008). Firstly, the “pickSoftThreshold” function determined the ideal weighted parameters for adjacent functions, which then served as soft thresholds in the construction of the network. Following this, a weighted adjacency matrix was established. To categorize gene modules, hierarchical clustering was employed, leveraging a topological overlap matrix (TOM) and employing a dissimilarity index (1-TOM). The subsequent step involved evaluating the association of each gene module with the specific disease, identifying the module with the strongest correlation as the key module. Genes within this key module were then earmarked for further investigation.

Enrichment analyses of key module genes

The key module genes identified by WGCNA were subjected to Gene Ontology Biological Process (GO_BP), Kyoto Encyclopedia of Genes and Genomes (KEGG) and Reactome enrichment analyses via the Database for Annotation, Visualization and Integrated Discovery (DAVID; https://david.ncifcrf.gov/) (Sherman et al., 2022). The results of the enrichment analysis were visualized by plotting dot plots using the “ggplot2” R package, exhibiting the top 10 most significantly enriched terms in each category.

Differential expression analysis

Differential expression analysis was performed using the “limma” R package (Ritchie et al., 2015). Subsequently, differential expression analysis was performed using “lmFit”, “eBayes”, and “topTable” functions. Genes with adjusted P-value (false discovery rate (FDR)) < 0.05 and —log2 fold change— (—logFC—) > 0.5 were identified as differentially expressed genes (DEGs). Finally, the DEGs were visualized using the “ggplot2” R package and the “pheatmap” R package to respectively generate volcano maps and heat maps.

Construction of protein-protein interaction networks for DEGs

The protein-protein interactions (PPI) network of DEGs was constructed through the STRING online platform (https://david.ncifcrf.gov/) (Szklarczyk et al., 2023). Subsequently, the PPI network was imported into Cytoscape software and screened for hub genes based on the cytoHubba plugin (Chin et al., 2014). A total of four diverse cytoHubba algorithms, namely Degree, EPC, MCC, and MNC were used for screening hub genes.

Identification of sepsis-related key genes

Using the “UpSetR” R-package and the “VennDiagram” R-package, the intersecting genes of both the genes in the key modules identified by WGCNA and the hub genes under the four algorithms were obtained as the sepsis-related key genes.

Validation of key genes

Validation of key genes was performed in an independent dataset, GSE131761 (containing 81 sepsis cases and 33 critical controls; Table S1). Specifically, the ability of each key gene to distinguish among sepsis cases and critical controls was first identified in the GSE131761 dataset using the “pROC” R package with a receiver operating characteristic curve (ROC). Subsequently, the genes in the WGCNA key module were defined as the gene set for sepsis signature and the sepsis signature level of each sample in the GSE131761 dataset was quantified using the “ssgsea” algorithm of the “GSVA” R package. Finally, based on the results of the normality test, the differences in sepsis signature levels and key gene levels between sepsis cases and critical controls were assessed using the t-test or Wilcoxon test.

Immune cell analysis

The relative proportion of immune cells for each sample in the combined dataset was assessed using the “CIBERSORT” algorithm (Chen et al., 2018). Subsequently, based on the results of the normality test, a t-test or Wilcoxon test was used to compare the differences of each immune cell between sepsis cases and critical controls. Subsequently, based on the results of the normality test, Pearson or Spearman correlation analyses were performed to assess the correlation of each key gene with the immune cells. The results of the correlation analysis were visualized using the heatmap.

Validation in sepsis patients

To further corroborate the effectiveness of the key genes, we recruited 22 sepsis patients and 10 non-sepsis critically ill controls from the intensive care units (ICUs) in our hospital between September 2020 and December 2021. All the participants were 18 years or older and had an ASA classification of III-IV. Sepsis patients were included when they met the diagnostic criteria for sepsis (Singer et al., 2016) and had a clear focus of infection in the urinary (obstructive urinary tract disease), gastrointestinal (gastrointestinal tract perforation) or biliary tracts (acute purulent obstructive cholangitis). Non-sepsis critically ill controls were patients with multiple traumatic injuries of similar age to those with sepsis and without co-infection. Patients were excluded if they received antibiotics or hormones before ICU admission; were currently receiving renal replacement therapy; had any cancers, autoimmune diseases, cirrhosis, uremia or had recent history of blood transfusion. Comparison of basic information for patients with sepsis and non-sepsis critical illness was shown in Table S2.

All blood samples were collected within 24 hours of ICU admission and prior to emergency surgery, after which PBMCs were isolated. RNA was then extracted from the PBMCs with TRIzol reagent (12183-555; Invitrogen) according to the manufacturer’s instructions. cDNA was synthesized using the SuperScript™ III First-Strand Synthesis SuperMix (11752-050; Invitrogen) for quantitative real-time polymerase chain reaction (qRT-PCR). The qRT-PCR was performed utilizing the HiScript II Q RT SuperMix for qPCR (Vazyme, Nanjing, China) in a LightCycler 480 PCR system (Hoffmann-La Roche Ltd., Shanghai, China). The relative mRNA expression levels were computed through the application of 2 - ΔΔCT method, with GAPDH serving as an internal reference. The primer pairs used in the experimental procedure are listed in Table S3. The data was presented as mean ± SD and analyzed via the Student’s t-test. Statistical significance was established when the p value was less than 0.05.

Our study was approved by the Institutional Review Board of the Third Xiangya Hospital (2020-S373) in strict compliance with the Declaration of Helsinki. Furthermore, written informed consents were procured from all patients or their legally designated representatives in our study.

Result

Dataset combination and DEGs identification

The overall workflow of this study is illustrated in Fig. 1. Principal Component Analysis (PCA) indicated that samples from GSE28750 and GSE74224 were batch-effect corrected and formed a new combined dataset (Figs. 2A–2B). Subsequently, in the combined dataset, a total of 293 DEGs (including 173 upregulated DEGs and 120 downregulated DEGs) were identified between sepsis and critical control (Figs. 2C–2D).

Figure 1 Overall flowchart of the study.

Figure 2 Dataset Combination and DEGs Identification.

(A) PCA shows the distribution of samples before batch effect removal between GSE28750 and GSE74224. (B) PCA shows the distribution of samples after batch effect removal between GSE28750 and GSE74224. (C) Volcano plot of 293 DEGs identified in the combined dataset, where red dots represent 173 upregulated DEGs, and blue dots represent 120 downregulated DEGs. (D) Heatmap of 293 DEGs identified in the combined dataset, where red squares represent 173 upregulated DEGs, and blue squares represent 120 downregulated DEGs.

WGCNA and identification of key module

First, we opted for a β value of 5, corresponding to a scale-free R2 of 0.85, to establish scale-free networks (Fig. 3A). After merging similar modules, a total of 9 gene modules were identified in the combined dataset (Fig. 3B). Correlation analysis revealed that the greenyellow module was significantly correlated with sepsis (correlation coefficient: 0.7, P = 2e−19) (Fig. 3C). Genes within the greenyellow module will be used for subsequent analysis. Subsequently, correlation analysis between module membership and gene significance revealed that these genes were highly correlated with both module and phenotype (cor = 0.64, p = 2.3e−128) (Fig. 3D).

Figure 3 WGCNA and identification of key module.

(A) Selection of soft threshold for WGCNA to ensure scale independence and mean connectivity. (B) The combined dataset underwent hierarchical clustering based on a topological overlap matrix (1-TOM), resulting in a dendrogram encompassing all genes. Each branch within the clustering tree signifies a gene, with co-expression modules delineated by distinct colors. (C) Correlation of various gene modules with sepsis. (D) Correlation analysis between module membership and gene significance.

Enrichment analysis of greenyellow module genes

The greenyellow module genes were subjected to enrichment analysis through the DAVID platform, and the results were presented in dot plots (Fig. 4). Specifically, upon conducting GO BP enrichment analysis, it was observed that genes within the greenyellow module showed significant enrichments in processes such as apoptotic process, cell division, intracellular signal transduction, protein phosphorylation, and inflammatory response (Fig. 4A). As for the KEGG pathway enrichment analysis, these genes exhibited notable associations with pathways including MAPK signaling pathway, prion disease, diabetic cardiomyopathy, chemical carcinogenesis—reactive oxygen species, and cell cycle (Fig. 4B). Moreover, analysis of Reactome pathways highlighted significant enrichments encompassing immune system, innate immune system, cell cycle, and neutrophil degranulation among the genes within the greenyellow module (Fig. 4C).

Figure 4 Results of enrichment analysis.

(A) Top 10 enriched terms identified by GO BP enrichment analysis. (B) Top 10 enriched terms identified by KEGG pathway enrichment analysis. (C) Top 10 enriched terms identified by Reactome pathway enrichment analysis.

Construction of PPIs for DEGs and identification of hub genes

Through the STRING platform, a PPI network of DEGs was constructed, containing 232 nodes and 821 edges (Fig. 5A). The PPI network included 143 up-regulated DEGs and 89 down-regulated DEGs (Table S4). In addition, based on four different algorithms, namely Degree, EPC, MCC, and MNC, top 20 hub genes in the PPI network were identified for subsequent analysis (Fig. 5B).

Figure 5 Construction of PPIs for DEGs and identification of hub genes.

(A) PPI network of DEGs generated by the STRING platform, containing 232 nodes and 821 edges. Red nodes represent up-regulated DEGs and blue nodes represent down-regulated DEGs. (B) PPI hub genes identified by four diverse algorithms (Degree, EPC, MCC, and MNC).

Identification and validation of sepsis key genes

Upset plot and Venn diagram showed that LTF, LCN2, ELANE, MPO, and CEACAM8 existed both in the greenyellow module and in the top 20 hub genes under the four algorithms (Fig. 6A). In the external validation dataset GSE131761, the ROC demonstrated that five key genes exhibited a promising ability to differentiate sepsis patients from critical controls (Fig. 6B). Using the “ssGSEA” algorithm, sepsis signature scores were found to be significantly higher in sepsis patients than in critical controls in the GSE131761 dataset, thus demonstrating the reliability of the greenyellow module identified by the WGCNA (Fig. 6C). In addition, the expression levels of LTF, LCN, ELANE, MPO, and CEACAM8 were all up-regulated in the peripheral blood of sepsis patients than critical controls in the GSE131761 dataset (Fig. 6D), which is consistent with the previous findings.

Figure 6 Identification and validation of sepsis key genes in microarrays.

(A) Upset plot and Venn diagram showed the sepsis key genes from the intersections of the top 20 genes of all four algorithms by CytoHubba and greenyellow module genes of the WGCNA. (B) ROC curve analysis of LTF, LCN, ELANE, MPO, and CEACAM8 in the GSE131761 dataset. (C) The greenyellow module genes in the WGCNA were defined as the gene set for sepsis signature, and the sepsis signature level of each sample in the GSE131761 dataset was quantified using the “ssgsea” algorithm. The differences in sepsis signature levels (ssGSEA score) between sepsis patients and critical controls were assessed. (D) The mRNA expression levels of LTF, LCN, ELANE, MPO, and CEACAM8 in sepsis patients and critical controls of GSE131761 dataset.

Subsequently, based on an external validation cohort from our hospital, qRT-PCR demonstrated significant up-regulation on LTF, LCN2, ELANE, MPO, and CEACAM8 in peripheral blood of sepsis patients versus controls (Fig. 7), thus further enhancing the reliability of the findings.

Figure 7 Validation of the key genes in sepsis patients recruited in our hospital.

The relative expression of LTF (A), LCN (B), ELANE (C), MPO (D), and CEACAM8 (E) between sepsis patients and non-sepsis critical controls by qRT-PCR. Data were expressed as mean ± SD. * p < 0.05; ** p < 0.01; *** p < 0.001.

Immune cell analysis

Figure 8A shows the distribution of circulating immune cells in the combined dataset of 84 sepsis patients and 42 critical controls. Figure 8B shows that circulating monocytes were significantly upregulated in sepsis patients compared to critical controls, whereas mast cells resting and T cells CD8 were significantly downregulated compared to critical controls. Correlations between immune cells and five key genes were assessed in patients with sepsis, revealing that plasma cells, macrophages M0, monocytes, T cells regulatory (Tregs), eosinophils and NK cells resting were simultaneously and significantly associated with more than two key genes (Fig. 8C).

Figure 8 Results of immune cell analysis.

(A) Histogram of the proportion of each type of circulating immune cell of 84 patient with sepsis and 42 critical controls in the combined dataset. (B) Boxplot of the relative expression of each immune cell subtype between patient with sepsis and critical controls. (C) Heatmap showing the correlation between immune cells and five key genes. * p < 0.05; ** p < 0.01.

Discussion

For critically ill patients in the intensive care unit, early and accurate diagnosis of sepsis is essential to reduce the morbidity and mortality. Once sepsis has progressed to the advanced stage of septic shock, the problem has become so severe that it may be too late for treatment. Hence, there is an urgent need to identify potential biomarkers that could aid in improving the diagnosis and prognosis of sepsis. In the present study, we utilized an integrative approach to identify 293 DEGs that are associated with sepsis from the microarray datasets GSE28750 and GSE74224. Notably, we identified five key genes (LTF, LCN2, ELANE, MPO, CEACAM8) associated with sepsis using the intersections of the top genes identified by cytoHubba and greenyellow module genes of the WGCNA. Furthermore, ROC curves were conducted to demonstrate good specificity of these key genes for sepsis diagnosis. These key overlapping genes were validated in an independent external dataset (GSE131761) and sepsis patients recruited in our hospital. Our novel findings bear significant implications for the clinical management of sepsis and offer promising avenues for the development of new diagnostic and therapeutic strategies.

Our investigation has identified five key genes, namely LTF, LCN2, ELANE, MPO and CEACAM8. Microarray analysis and RT-qPCR experiments revealed a significant up-regulation of these critical genes in sepsis patients. The protein products encoded by LTF and LCN2 genes were prominently found in the secondary granules of neutrophils and are considered innate immune proteins to fight infection (Huang et al., 2022; Zhao et al., 2018). LTF protein is a potent regulator of inflammatory homeostasis through its ability to reduce C-reactive protein levels (Pammi & Suresh, 2020). A number of studies have found that LCN-2 expression is upregulated in sepsis (Lu et al., 2019; Mertens et al., 2020). LCN2 is involved in the regulation of inflammatory processes, particularly in neutrophil activation and migration (Lu et al., 2019). Previous study showed that LTF and LCN2, as differentially expressed genes, provide a reliable tool to differentiate septic shock from non-septic shock in postoperative patients (Martínez-Paz et al., 2021). Similarly, our investigation revealed that the expression levels of LTF and LCN2 were significantly increased in sepsis patients when compared with critical controls, implicating a potential link between LTF, LCN2 and sepsis. ROC curve analysis also suggested that LTF and LCN2 may serve as valuable biomarkers to differentiate sepsis patients from those with non-septic inflammatory conditions. Further research is warranted to elucidate the precise mechanism underlying their involvement in sepsis and to explore the therapeutic potential.

ELANE is a neutrophil elastase and plays an important role in pathogen killing (Zhao et al., 2023). However, its over-activation may lead to neutrophil death and cause tissue destruction, which is associated with a poor prognosis in sepsis (Kambara et al., 2018). Previous studies found that ELANE was overexpressed in patients with sepsis and correlated with the severity of sepsis (Baghela et al., 2022; Zhang et al., 2022; Zhang et al., 2020). Consistently, in our study, ELANE was found to be expressed at significantly higher levels in sepsis patients compared to non-sepsis critically ill patients. MPO (Myeloperoxidase) is an enzyme produced by neutrophils and is an important component of the innate immune system (Lin et al., 2024). MPO is involved in the bactericidal activity of neutrophils and is released when neutrophils are activated to remove the source of infection (Lin et al., 2024). According to our findings, sepsis patients had higher levels of MPO expression in peripheral blood mononuclear cells, which is consistent with other studies (Schrijver et al., 2017), suggesting that MPO has the potential to serve as a biomarker to distinguish sepsis from infection-negative systemic inflammation. However, the exact role of MPO in sepsis still requires further research.

CEACAM8, also known as CD66b, is a protein of the CEACAM family that is mainly expressed on the surface of neutrophils (Ribon et al., 2019). This protein plays a key role in neutrophil adhesion and migration. In septic conditions, CEACAM8 is overexpressed, prompting neutrophils to localize to the site of infection, release inflammatory mediators and work to clear pathogens (Gray-Owen & Blumberg, 2006; Schmidt et al., 2012). Our findings are in line with this phenomenon, showing that CEACAM8 expression is significantly upregulated in patients with sepsis. Therefore, this molecule may be considered as a potential marker of sepsis and could be used to assess the severity of infection and inflammation.

In addition, immune cell infiltration in sepsis and critical control samples was analyzed using CIBERSORT. Distinct disparities were observed in the infiltration patterns of four immune cells types between the two groups: compared to critical control group, sepsis group had higher proportions of monocytes and activated memory CD4+ T cells, and lower proportions of resting mast cells and CD8+ T cells. In the early stage of sepsis, monocytes play a critical role in orchestrating the host immune response by detecting pathogens, performing phagocytosis and killing of pathogens, regulating the immune response, and modulating effector cell functions (Crouser et al., 2019). During sepsis, mast cells can be activated and release inflammatory mediators, leading to inflammatory response and microcirculatory disturbance (Nautiyal et al., 2009; Udayanga et al., 2016). It is worth mentioning that these innate immune cells may also serve as protective elements against sepsis during the initial phases of infection (Udayanga et al., 2016). Previous literatures have suggested that activated memory CD4+ T cells are involved in modulating the function of neutrophils, fostering bactericidal functions, and improving animal survival during sepsis (Taylor et al., 2020). In addition, studies have shown that sepsis is associated with decreased counts of CD8+ T cells, which are essential for combating infections and maintaining immune function (Guo et al., 2021). These findings underscore the importance of immune cell infiltration in comprehending the pathogenesis and classification of sepsis.

Enrichment analysis of the key module genes identified by WGCNA indicated that immune pathways (innate immune system and neutrophil degranulation), inflammatory pathways (MAPK signaling pathway) and apoptotic process participated in the pathology of sepsis. In addition, we considered the key module genes as a potential signature for sepsis in critically ill patients with ssGSEA method. The ssGSEA score was significantly higher in sepsis group compared with non-septic critically ill controls in the validation set, suggesting that this key module has some extent of diagnostic robustness.

With regard to qPCR analysis of peripheral blood mononuclear cell-related genes, it may take 3 h or less to complete, allowing rapid clinical integration for early diagnosis and treatment based on the patient’s clinical condition (Pandey et al., 2024). Changes in the expression levels of these genes can aid in the early diagnosis of sepsis in patients with severe diseases and the prompt treatment to reduce progression and mortality rates (Gotts & Matthay, 2016). Monitoring the changes in these biomarkers can also help to assess disease progression and treatment outcomes, predict prognosis and adjust treatment options (Gotts & Matthay, 2016). Patients with significant clinical symptoms, suspected sepsis, and requiring urgent treatment may be preferred subjects for research.

Conclusion

In summary, our investigation conducted comprehensive bioinformatics analysis on two gene datasets (GSE28750 and GSE74224) and discerned LTF, LCN2, ELANE, MPO and CEACAM8 as key up-regulated genes in sepsis patients when compared with non-sepsis critically ill controls. Furthermore, we confirmed the expression of these five key genes in external independent datasets and septic patients recruited in our study. These findings hold promise in advancing the exploration of potential diagnostic biomarkers and molecular mechanisms for sepsis.

Supplemental Information

Supplemental Information 1 Overview of sepsis datasets used in this study

Supplemental Information 2 Characteristics of 22 Sepsis and 10 Non-sepsis patients

Supplemental Information 3 The sequences of siRNAs used in the study

Supplemental Information 4 The PPI network included 143 up-regulated DEGs and 89 down-regulated DEGs

Supplemental Information 5 Author justification

Additional Information and Declarations

Competing Interests

Author Contributions

Human Ethics

Data Availability

The authors declare there are no competing interests.

Guoxin Lin conceived and designed the experiments, performed the experiments, analyzed the data, authored or reviewed drafts of the article, and approved the final draft.

Nannan Li conceived and designed the experiments, performed the experiments, analyzed the data, authored or reviewed drafts of the article, and approved the final draft.

Jishi Liu conceived and designed the experiments, prepared figures and/or tables, authored or reviewed drafts of the article, and approved the final draft.

Jian Sun conceived and designed the experiments, prepared figures and/or tables, and approved the final draft.

Hao Zhang conceived and designed the experiments, prepared figures and/or tables, authored or reviewed drafts of the article, and approved the final draft.

Ming Gui conceived and designed the experiments, prepared figures and/or tables, and approved the final draft.

Youjie Zeng conceived and designed the experiments, analyzed the data, authored or reviewed drafts of the article, and approved the final draft.

Juan Tang conceived and designed the experiments, analyzed the data, prepared figures and/or tables, authored or reviewed drafts of the article, and approved the final draft.

The following information was supplied relating to ethical approvals (i.e., approving body and any reference numbers):

Third Xiangya Hospital of Central South University approval to carry out the study within its facilities (Ethical Application (2020-S373).

The following information was supplied regarding data availability:

The raw data are available in the Supplementary Files.

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
