# Peer review of "Identification of key genes as potential diagnostic biomarkers in sepsis by bioinformatics analysis"

_PeerJ, doi:10.7717/peerj.17542_

## Round 0.1 · original submission · Major Revisions

All three reviewers have given suggestions for revision, and the author is requested to revise them carefully.

Reviewer 1 ·

Basic reporting

The manuscript requires a thorough proof-reading to avoid recurring typos, inaccurate grammatical assertions.

Experimental design

Previously published work seems to have adequately established both FYN and CD247 as potential diagnostic biomarkers for sepsis (10.1155/2020/3432587, 10.2174/1386207324666210816123508). Request the authors to clearly establish the knowledge gap addressed in this study.

The approach used to combine microarray datasets across platforms prior to performing differential gene expression analysis seems elusive and runs the risk of inheriting platform related confounders that can negatively influence the outcome (Fig 1B-D).
Request the authors to provide a detailed explanation of the normalisation step with adequate literature evidence. Alternatively, performing a meta-analysis seems like a more optimal approach in this instance and can side-step the complexity associated with combining these datasets.

Validity of the findings

This work is largely a derivative of existing work (10.1155/2020/3432587, 10.2174/1386207324666210816123508).

·

Basic reporting

See point 2.

Experimental design

The study appears to have a two-stage design:
1. First, the authors used three publically available gene expression data sets to conduct their discovery analysis. In total these included 309 patients with sepsis and 123 patients with criticall ilnesses.
- Very limited clinical information of the cohorts are provided, if they are available please create a table with these characteristics. Which countries? Can the authors justify to combine these cohorts?
- Definitions of sepsis and criticall illnesses are missing. Are they comparable between the cohorts?
- A differentially expresssed genes (DEGs) analysis was done, but it inclear between which groups (sepsis versus criticall ilness? healthy controls?).

2. The identified DEGs are "validated" in 25 patients with sepsis and 16 non-sepsis critically ill controls.
- Also for these patients there is extremely limited information available. What kinds of patients? Post-operative sepsis? community acquired infections? Also for the non-sepsis critically ill patients. Definitions?
- How are these patients selected? From which time period? Inclusion and exclusion criteria?

In its current form, the manuscript is insufficient to comment on the results and conclusions. I would recommend to rewrite the manuscript by using a suitable guideline for reporting standards of observational studies.

Validity of the findings

See point 2.

Additional comments

See point 2.

·

Basic reporting

1) The manuscript is well-written and easy to understand. The authors use clear and concise language, and is also professionally formatted adhering to standard conventions.
3) Please add the expansion for MAD in the methods section (line 74)
4) "Subsequently, the PPI hub genes were intersected with the key module genes of the WGCNA and tissue-specific genes by using the Venn diagram." - is this to identify the common genes found in all for algorithms, black module and tissue specific genes? If yes, then including that venn diagram in figure 3 might help better understand.
5) Color coding in Figure 3 B needs an explanation.
6) Replace "Where" with ", with the" to make the sentence more coherent in line 84.
7) "Informed consents were collected from." in line 128 could be removed since its redundant and incomplete.
8) In the PPI network, absolute value of PCC>0.4 might include low confidence also. Why not use 0.7 and select only genes with high confidence if the result still continues to cover the list of gene identified through overlap.

Experimental design

The authors proposed a novel approach and performed a rigorous investigation using variety of methods to identify and validate the key genes, and they provide sufficient detail in the methods section to allow other researchers to replicate their work.

Validity of the findings

No comment

---

## Round 0.2 · Major Revisions

The author should carefully modify the manuscript according to the reviewer's requirements and answer the reviewer's questions.

Reviewer 1 ·

Basic reporting

Appreciate the authors for improving the readability of the manuscript.

Experimental design

Appreciate the authors showcasing the knowledge gap, however it is not reflected in the introductory sections leading to confusion regarding the control group.
Request the authors to explicate the experimental design in the introductory section for better readability.

Please see section specific comments to improve the readability and impact of the manuscript.

Section 2.2
Combat and limma::removeBatcheffect are two different methods for batch effect removal. The authors' have used of both combat and limma::removeBatcheffect for batch effect removal.
Kindly clarify.

Section 3.1
Log Fold change threshold (+/- 0.3785) seem arbitrary, kindly provide rationale for the same.

Section 3.6
Request authors to use Upset plot (UpSetR; R package) to improve the depiction of intersections in Fig.5A.

The authors have performed a detailed WGCNA analysis for module discovery and would benefit the readership to see it followed up with functional analysis (KEGG, GO etc) of these modules.
The utilisation of Venn diagram approach may be too stringent and may risk losing true biological signal in the WGCNA module related to sepsis phenotype.
In addition to the venn diagram approach, request the authors to also explore the black module genes as a potential signature for sepsis in critically ill patients (Ref: ssgsea method in R package GSVA, https://bioconductor.org/packages/devel/bioc/vignettes/GSVA/inst/doc/GSVA.html).
The comparison of ssgsea scores in sepsis vs non-sepsis patients in the validation set can be useful in assessing the robustness of the modules discovered by WGCNA.

Validity of the findings

Authors have clarified the key rationale and its broader impact.

Additional comments

No comments

·

Basic reporting

Overall, the document has been significantly improved with the adjustments, however it needs some more adjustments.

Experimental design

1. - The authors argue that the cohorts are comparable, however, the study by Martinez-paz is solely done in post-operative patients, and the patients in the study by Scicluna et al. were more diverse (pneumonia, abdominal etc). These and other patient characteristics are not included in the study. How do the authors justify to combine these vastly different cohorts in their study?

-Please also add the definitions used for non-sepsis critical ilness by the original studies. "mirrored" is a vague, non-scientific term.

2. The authors provided more information on their validation cohorts. However, it is still limited. What kind of gastrointestinal source? Appendicitis? Post-operative? Comorbidities? ASA-classification?

Validity of the findings

The paper seems to confirm what was already established in the literature; the role of FYN and CD247 in sepsis. The authors imply that these targets can be used for biomakrers for early diagnosis and treatment. What is the authors view on implementation of these results in clinical practise? How could it help in treatment? Which patients would you select for this biomarker? It takes days/weeks to analyse gene expression data, how can it be employed in practise?

---

## Round 0.3 · Minor Revisions

The authors are requested to carefully revise the manuscript and answer the questions raised by the reviewers.

Reviewer 1 ·

Basic reporting

No comment

Experimental design

No comment

Validity of the findings

The newly added analysis method described in section 2.9 is unclear and needs to adequately supported by literature evidence. Using single gene to perform GSEA is conceptually problematic and encourage the authors to review the principles of GSEA (https://www.pnas.org/doi/10.1073/pnas.0506580102) prior to using a single gene to predict pathway related effects.
Recommend exclusion of all the sections/figures (Section 2.9, 3.7 and Fig. 9) related to this analysis from the final version of the article. Fig.1 and Discussion section should also be amended to reflect these changes.

Additional comments

Great to see the fantastic effort by authors to have robustly addressed the major suggestions/changes.

Appreciate the authors for adding Figure 1, it has greatly improved the readability of the article.

Minor suggestion for amending Figure 1.
The arrow between boxes "Hub genes identified by CytoHubba" and "Key genes" should be removed. The "key genes" were identified sequentially following the intersection of genes identified by both CytoHubb and WGCNA.

Minor suggestions for amending Figure 6.
Combine Figure 6A-B.
Combine all violin plots for single genes 6E-I into one figure.

·

Basic reporting

The authors have adequately adressed my concerns. Please make sure that all answers in the rebuttal letter are integrated in the main text. For example, it seems that the last answer is not incorporated in the Discussion section.

Experimental design

NA

Validity of the findings

NA

Additional comments

NA

---

## Round 0.4 · accepted · Accept

After revisions, two reviewers agreed to publish the manuscript. I also reviewed the manuscript and found no obvious risks to publication. Therefore, I also approved the publication of this manuscript.

Reviewer 1 ·

Basic reporting

No comment

Experimental design

No comment

Validity of the findings

No comment